# Polymyxin B in Combination with Glycerol Monolaurate Exerts Synergistic Killing against Gram-Negative Pathogens

**DOI:** 10.3390/pathogens11080874

**Published:** 2022-08-02

**Authors:** Yun Zheng, Ning Yang, Yuting Ding, Jiajia Li, Yanyan Liu, Haoran Chen, Jiabin Li

**Affiliations:** 1Department of Infectious Disease, The First Affiliated Hospital of Anhui Medical University, Hefei 230022, China; zhengyun2022@126.com (Y.Z.); ning_y21@126.com (N.Y.); 2045010947@stu.ahmu.edu.cn (Y.D.); liuyanyan725@ahmu.edu.cn (Y.L.); 2Department of Emergency, The First Affiliated Hospital of Anhui Medical University, Hefei 230022, China; 3The Center for Scientific Research, The First Affiliated Hospital of Anhui Medical University, Hefei 230022, China; mud_loverr@126.com; 4Anhui Center for Surveillance of Bacterial Resistance, Hefei 230022, China; 5Institute of Bacterial Resistance, Anhui Medical University, Hefei 230022, China

**Keywords:** polymyxin B (PMNB), glycerol monolaurate (GML), synergistic antimicrobial activity, multidrug-resistant (MDR), reactive oxygen species (ROS), Gram-negative pathogens, pulmonary bacterial infection

## Abstract

The rapid emergence and spread of multidrug-resistant (MDR) bacterial pathogens pose a serious danger to worldwide human health, and resistance to last-resort drugs, such as polymyxins, is being increasingly detected in MDR Gram-negative pathogens. There is an urgent need to find and optimize combination therapies as an alternative therapeutic strategy, with a dry pipeline in novel antibiotic research and development. We found a monoester formed from the combination of lauric acid and glycerol, glycerol monolaurate (GML), possessing prominent antibacterial and anti-inflammatory activity. However, it is still unclear whether GML in combination could increase antimicrobial activity. Here, we reported that polymyxin B (PMNB) combined with GML exhibited a synergistic antimicrobial impact on Gram-negative strains in vitro, including clinical MDR isolates. This synergistic antimicrobial activity correlated with the destruction of bacterial cell structures, eradication of preformed biofilms, and increased reactive oxygen species (ROS) accumulation. We also showed that PMNB synergized with GML effectively eliminated pathogens from bacterial pneumonia caused by *Klebsiella pneumoniae* to rescue mice. Our research demonstrated that the PMNB and GML combination induced synergistic antimicrobial activity for Gram-negative pathogens in vitro and in vivo. These findings are of great importance for treating bacterial infections and managing the spread of infectious diseases.

## 1. Introduction

The emergence of antimicrobial resistance (AMR) and the consequent prevalence of multidrug-resistant (MDR) Gram-negative bacteria represent a considerable and increasing threat to public health. In particular, *Enterobacteriaceae*, such as pathogenic *Escherichia coli*, *Klebsiella pneumoniae*, *Acinetobacter baumannii*, and *Pseudomonas aeruginosa*, are important pathogens causing both nosocomial and community-acquired infections [1,2]. With the widespread overuse of antibiotics in recent years, these bacteria are capable of resisting antibiotics via multidrug efflux pumps, porins, and the impermeable outer membrane. To cope with their challenging environments, bacteria have now evolved various adaptive resistance elements that cover all classical pathways targeted by conventional antibiotics: cell walls, proteins, RNA, and DNA synthesis [3]. Regrettably, the discovery and development of novel antimicrobial agents are limited by long development cycles, high cost, and low clinical trial success rate, which could lead to a lack of effective drugs against these “superbugs” in the near future. Therefore, there has been renewed interest in reviving older antibiotics that were deemed to have a very narrow therapeutic index, with toxicity in clinical use [4]. Meanwhile, the elimination of bacterial biofilms is also a promising strategy for palliating the appearance of AMR.

Taking into account directly targeting and disrupting the bacterial cell membrane can potentially circumvent many resistance mechanisms; polymyxins, a class of membrane-targeting antibiotics, have tremendous potential that is yet to be fully exploited. Polymyxins are antibacterial peptides produced by the Gram-positive bacterium *Paenibacillus polymyxa* that include polymyxins A–E, of which only polymyxins B and E (also known as colistin) are used clinically [5]. They compete with magnesium and calcium ions in lipid A binding and are subsequently inserted into the phospholipid bilayer, disrupting the structural integrity of the membrane and promoting other polymyxin molecules to pass through the outer membrane, or act via vesicle–vesicle contact to disrupt the phospholipid composition, and then induce cytoplasmic outflow, thus leading to cell death [6]. Polymyxins were abandoned in the 1970s because of their potential nephrotoxicity, but they have recently emerged as a last-resort defence against Gram-negative strains resistant to all other currently available antibiotics. However, the therapeutic effect of polymyxins has been greatly weakened by the emergence of the plasmid-mediated resistance to polymyxin (MCR-1) in *Enterobacteriaceae* [7,8]. Thus, efforts have been directed at improving the therapeutic efficacy of polymyxins in order to save lives from the dangerous infections caused by MDR Gram-negative pathogens.

Several recent publications demonstrated that it is very important to control the growth of MDR Gram-negative pathogens, increase local bioavailability, and reduce systemic toxicity through improving the therapeutic characteristics of polymyxin, such as by nanotechnology-based modification and combination therapy [9,10,11,12]. Meanwhile, rational combinations or novel therapeutic approaches targeting the infection and inflammation through bioactive dietary components are receiving growing attention [13]. Various free fatty acids (FAs) and their monoglyceride derivatives have long been found to exhibit antimicrobial activity against numerous bacterial pathogens [14]. Glycerol monolaurate (GML), a mild surfactant formed by glycerol and lauric acid (Appendix A), exists naturally in coconut oil and human milk, is considered as a preservative and emulsifier in food, and is generally recognized as safe (GRAS) for oral use by the FDA [15]. Although it is not currently in clinical use, GML has potent antimicrobial activity against enveloped viruses and most Gram-positive bacteria, such as *Streptococci* and *Staphylococci* [16]. Unfortunately, it has only a limited effect on Gram-negative bacteria due to the presence of the intact lipopolysaccharide (LPS) [17]. This problem has led to the search for an antibacterial drug combination, which could destabilize the external structure of Gram-negative bacteria for achieving the desired therapeutic effect.

Considering polymyxin B (PMNB) has lower nephrotoxicity than colistin and can achieve the desired therapeutic effect at a lower concentration to dilute serious side effects, such as nephrotoxicity, the following studies sought to determine new PMNB combination therapies against MDR Gram-negative bacterial infections and confirm the mechanistic causes for their antimicrobial activities [18,19]. In addition, because bactericidal antibiotics are known to be related to the reactive oxygen species (ROS) produced, which will lead to bacterial cell death, we hypothesized that PMNB combined with GML reconstructed the antibacterial activity against Gram-negative bacteria and eliminated MDR isolates via ROS [20,21]. The present work provides new insights into the treatment of Gram-negative pathogens infections, highlighting the role of PMNB synergized with GML in enhancing pronounced antimicrobial effects in vitro and in vivo. We demonstrated that ROS generation, bacterial cell membrane integrity disruption, and biofilm inhibition are critical elements in synergistic antimicrobial activity by combining a scanning electron microscope (SEM), biofilm formation assay, and ROS assay. Furthermore, PMNB combined with GML enhanced the clearance of Gram-negative bacteria in pulmonary infections and recovered the antimicrobial activity of PMNB to MDR strains. This novel drug combination should be helpful for improving the treatment of MDR Gram-negative bacteria-induced infections in clinical work.

## 2. Results

### 2.1. Antimicrobial Susceptibility of PMNB and GML

The MICs of PMNB and GML were tested against various Gram-negative bacteria and are shown in Table 1. According to the Clinical and Laboratory Standards Institute (CLSI) and the European Committee on Antimicrobial Susceptibility Testing (EUCAST), two clinical MDR isolates, *A. baumannii* HR13 (MIC, 4 μg/mL) and *K. pneumoniae* KP024 (MIC, 8 μg/mL), were resistant to PMNB; the other strains had similar MIC values ranging from 0.5 to 1 μg/mL. However, GML did not exhibit any effective antibacterial activities at a concentration of up to 4000 μg/mL for all Gram-negative pathogens.

### 2.2. Synergistic Effect of PMNB Alongside GML against Gram-Negative Pathogens

By using checkerboard assays, the synergistic effect of PMNB and GML was tested and is presented in Table 1 and Figure 1. For antibiotic-sensitive Gram-negative strains, when in combination with GML (<1/8× MIC), a four-fold reduction was observed in the PMNB MIC value against *E. coli* BW25113, and the FICI was estimated to be less than 0.375, suggesting that PMNB and GML exerted a synergistic effect (Figure 1A). The treatment of ATCC 17978 resulted in the same FICI value (FICI < 0.375; Figure 1C). Similar results were also observed for ATCC 43816 (FICI< 0.25; Figure 1B) and PA14 (FICI < 0.3125; Figure 1D). Importantly, GML sensitized the MDR and polymyxin-resistant (PR) KP024 strain to PMNB by eight-fold, reducing the MIC values of GML to 500 μg/mL (FICI < 0.25, Figure 1E), and the MIC of HR13 to PMNB was also decreased by eight-fold in the presence of GML (FICI < 0.25, Figure 1F).

### 2.3. Antimicrobial Killing of PMNB in Combination with GML

We next investigated whether GML could increase the antimicrobial activity of PMNB in Gram-negative bacteria through time–kill assays (Figure 2 and Figure 3). PMNB (2× MIC), in combination of GML (500 μg/mL), rapidly increased ~2.1 log_10_ CFU/mL killing in *E. coli* BW25113 (Figure 2A), ~2 log_10_ CFU/mL killing in *K. pneumoniae* ATCC 43816 (Figure 2B), ~1.8 log_10_ CFU/mL killing in *A. baumannii* ATCC 17978 (Figure 2C), and ~2.6 log_10_ CFU/mL killing in *P. aeruginosa* PA14 (Figure 2D) within 90 min.

Meanwhile, in the presence of 500 μg/mL GML, 4× MIC PMNB exhibited more robust antimicrobial killing (Figure 2). Therefore, we further used 4× MIC PMNB to test the synergistic killing with or without GML within 24 h and observed stable antibacterial activity (Figure 3). This combination treatment resulted in strong ~3.8 log_10_ CFU/mL killing of *E. coli* BW25113 (Figure 3A), ~4.5 log_10_ CFU/mL killing of *K. pneumoniae* ATCC 43816 (Figure 3B), ~4.1 log_10_ CFU/mL killing of *A. baumannii* ATCC 17978 (Figure 3C), and ~4.8 log_10_ CFU/mL killing of *P. aeruginosa* PA14 (Figure 3D) compared with PMNB alone at 24 h.

Moreover, *K. pneumoniae* KP024 and *A. baumannii* HR13 were relatively insensitive to PMNB, and antimicrobial killing was also attenuated compared with standard strains. However, PMNB-mediated lethality (2× MIC) was strongly enhanced in the presence of GML, and the number of bacteria was at least 2 log_10_ CFU/mL lower than that under the monotherapy of PMNB at 90 min (Figure 2E,F). Similarly, the PMNB and GML combination therapy showed a powerful bactericidal effect, with ~4 log_10_ CFU/mL killing for KP024 (Figure 3E) and ~3.7 log_10_ CFU/mL killing for HR13 (Figure 3F) following 24 h incubation when compared with PMNB alone. GML alone did not alter bacterial growth (Figure 2 and Figure 3).

### 2.4. Combined PMNB and GML Treatment Aggravates Morphological Damage

To explore the morphological changes in all tested strains following treatment with PMNB, GML, or both, we performed SEM experiments. The images (Figure 4) depict intact bacterial cell wall structures without treatment and with GML alone. PMNB monotherapy caused some ruptures on the cell membrane for a small part of the bacteria. Notably, large-scale membrane disruptions with membrane blebbing, rough surfaces, shrinking, and cratering were observed for the PMNB and GML combination. For clinical MDR isolates (Appendix A), PMNB monotherapy induced slight damage to the membrane. However, there were severe membrane destructions in response to combination treatment.

### 2.5. PMNB Synergized with GML against Biofilm Formation

The combination of PMNB and GML was further detected for its efficacy in eradicating established biofilms using clinical MDR KP024 and HR13 isolates, as well as standard PA14 and BW25113 strains as models. A significant dose-dependent inhibitory effect on biofilms was found after the monotherapy of PMNB. For KP024, 45.4% of the biofilm was eradicated at 2× MIC PMNB, and 77.4% of the biofilm was eradicated at 4× MIC PMNB compared with no treatment, respectively. As expected, PMNB synergized with GML exhibited the strongest impact on biofilms, and the biofilm inhibition rates were 81.4% for 2× MIC PMNB alongside GML, and 87.3% for 4× MIC PMNB alongside GML, suggesting that this combination displayed the best biofilm inhibition for KP024 compared with PMNB alone at 24 h (Figure 5A,B). In addition, similar results were found in HR13 and PA14. For HR13, the biofilm inhibition rates were 89.6% in the 2× MIC PMNB and GML combination, and 91.7% in the 4× MIC PMNB and GML combination compared with untreated groups (Figure 5C,D). For PA14, the biofilm inhibition rates were 92.7% in the 2× MIC PMNB and GML combination, and 94.1% in the 4× MIC PMNB and GML combination compared with controls (Figure 5E,F). Meanwhile, a decrease in biofilm formation was seen as a result of the combination treatment of *E. coli* BW25113 (Appendix A; *p* > 0.05), but the results were not significant. Conversely, such biofilm inhibition disappeared with GML monotherapy (*p* > 0.05). Our data indicated that the high antimicrobial activity of the PMNB–GML combination was tightly associated with preventing biofilm formation.

### 2.6. PMNB–GML Combination Stimulates ROS Production

As ROS was recently recognized as a common bactericidal mechanism of antibiotic activity, we next examined whether GML stimulates PMNB-induced ROS production. The intracellular ROS levels were detected for *E. coli* BW25113 and *K. pneumoniae* ATCC 43816 by incubating with carboxy-H2DCFDA. PMNB monotherapy had a mild effect on fluorescence intensity, but PMNB combined with GML significantly stimulated ROS accumulation in BW25113 (Figure 6A). Similarly, visible increases were observed in the ROS levels in ATCC 43816 (Figure 6B).

Next, we explored the effect of PMNB in combination with GML on ROS-induced bacterial cell death. In agreement with the assay of killing kinetics, the combination treatment showed strong synergistic killing effects on BW25113 (Figure 6C) and ATCC 43816 (Figure 6D). However, this combination-mediated synergistic killing was sharply weakened after the addition of 5% DMSO for BW25113 (increased by 1.3 log_10_ CFU/mL; *p* < 0.05) (Figure 6C), and ATCC 43816 (increased by 1.2 log_10_ CFU/mL; *p* < 0.05) (Figure 6D). Likewise, DMSO monotherapy did not affect bacterial growth at this dose (Appendix A; *p* > 0.05). Overall, these results show that the PMNB and GML combination causes synergistic antimicrobial killing, which is dependent on increased levels of intracellular ROS.

### 2.7. PMNB–GML Combination Contributed to Clearance of K. pneumoniae from Lung Infection

To persuasively evaluate the antibacterial activity of PMNB and GML in vivo, we established a mouse model of lung bacterial infection caused by *K. pneumoniae* ATCC 43816 (Figure 7A). Infected mice were separated into four groups, and treated with PBS (control), PMNB, GML, or both drugs together. In the *K. pneumoniae* model, the survival rate of mice was 75% following PMNB monotherapy (Figure 7B) at 96 h after inoculation. In contrast, 93% of mice in the PMNB and GML combination group survived until the end of the experiment. (Figure 7B; *p* < 0.01). After treatment that lasted for 96 h, the body weight was recorded every 24 h, and that of the treated mice began to increase after 48 h, but there was an absence of a statistically significant change between the treatment groups (Figure 7C; *p* > 0.05).

To investigate bacterial clearance in vivo, the bacterial loads in the infected lung from each group at 3 days post-treatment were evaluated. The monotherapy of PMNB showed antimicrobial activity by eliminating ~1.5 log_10_ CFU in the lung, but there was a sharp reduction (~2.2 log_10_ CFU) in the number of ATCC 43816 in the presence of PMNB and GML combination (Figure 7D). Instead, there was no significant difference in the bacterial burdens between the PBS-treated groups and GML-treated groups. These results were consistent with our findings in vitro. Collectively, *K. pneumoniae*-induced lung infection could be improved by using the PMNB and GML combination treatment.

## 3. Discussion

Lower respiratory infections are a major cause of morbidity and mortality around the world, with Gram-negative bacteria accounting for a large portion of the pathogens that cause infections [22,23]. *Pseudomonas aeruginosa*, *Escherichia coli*, *Klebsiella pneumoniae,* or *Acinetobacter baumannii* have been identified as major causative pathogens of different respiratory tract infections [24]. In recent years, the extensive use of antibiotics has led to the emergence of multidrug-resistant (MDR) Gram-negative pathogens and the traditional antibiotics to be ineffective or of limited use, which constitutes a great hazard to respiratory health. Coupled with the lack of novel antibiotics in the development pipeline nowadays, a critical approach is to revitalize existing antibiotics using adjuvants [25]. Strains resistant to the last-line PMNB have become increasingly more frequent, and PMNB combination therapy is considered a viable clinical strategy to rescue the bacterial killing efficacy against pathogens resistant to monotherapy [26,27].

In the current study, we found that PMNB synergized with GML increased the antimicrobial activity of PMNB against Gram-negative pathogens. Furthermore, the PMNB and GML combination therapy showed activity against Gram-negative bacteria, including clinical MDR isolates. Our results strongly indicate that GML can enhance PMNB-mediated protection against both susceptible and resistant bacterial phenotypes in vitro. Notably, GML alone displayed high MICs and undetected lethality, indicating that it is inactive against *Enterobacteriaceae* and *P. aeruginosa*, consistent with previous reports [28]. These data also imply that the antibacterial activity of GML is also elevated with PMNB; thus, further investigation is required to definitively understand the mechanism(s) of this combination against Gram-negative bacteria.

As PMNB targets the outer membrane and greatly increases membrane permeability, it may lead to improved antimicrobial efficacy of GML. Therefore, we initiated cellular morphology studies of Gram-negative bacteria. After the PMNB and GML combination treatment, we observed bacterial structural changes ranging from rough surfaces to blebbing of cell membranes, large-area rupture of the membrane, and presence of cellular fragments when viewed using SEM, which were more severely affected than monotherapy. Our observation suggests that cell morphology and membrane integrity were seriously destroyed after this combination treatment; following which GML may actively enter the intracellular space to exert potent antimicrobial effects on Gram-negative pathogens. In addition, bacterial biofilm formation is related to the decrease in the effectiveness of antibiotics, and has been reported as the primary reason for drug-resistant bacteria [29]. Among these Gram-negative pathogens, especially *P. aeruginosa*, once they form biofilms to establish chronic infections, they further develop to MDR, making difficult-to-treat infections become life-threatening [30]. Considering this issue, we next directly visualized the effect of this combination treatment on biofilm formation. Not surprisingly, higher biofilm inhibition rates were shown with the PMNB and GML combination in Gram-negative bacteria including MDR isolates when compared with PMNB alone. Our results collectively indicate that the synergistic killing is closely associated with membrane disruption and biofilm eradication.

It is curious as to how the PMNB and GML combination induced synergistic antimicrobial activity after entering the bacterial cells. As previously reported, while each of the bactericidal drugs tested have different primary drug–target interactions, almost all classes of bactericidal antibiotics utilize a common mechanism of killing involving free-radical damage of iron–sulfur clusters, leading to the destabilization or leaching of ferrous iron that participates in the Fenton reaction, resulting in ROS-induced damage, ultimately causing cell death [20,21]. PMNB was also no exception to this [31]. However, the role of ROS is debatable in killing bacteria [32]. Considering those findings, we propose that GML might affect the activity of PMNB via this bactericidal mechanism. Indeed, a dramatic increase in the ROS levels was observed for the combination of PMNB and GML compared with the monotherapy treatment. Furthermore, the PMNB and GML combination-induced rapid killing was partially reversed by using DMSO to alleviate ROS production. Consistent with previous results, the bacterial lethality was impaired by reducing the ROS accumulation for PMNB monotherapy; however, in the absence of GML, PMNB exhibited a minor increase in the level of intracellular ROS at the time of detection. It is plausible that ROS could make a great contribution to PMNB-induced killing and change with time. Taken together, these data demonstrated that the increased ROS generation, grievous bacterial cell structure destruction, and considerable inhibition of biofilm collaboratively contributed to the synergistic interactions between PMNB and GML.

Pneumonia is a serious and potentially life-threatening infectious disease affecting children and elderly people, the pathogenesis of which is involved in microbial access to the lower respiratory tract via community or hospital admission [33]. It may occur through the inhalation of the causal pathogen. Most instances of hospital-acquired pneumonia are caused by Gram-negative pathogens, represented by *K. pneumoniae* [34]. As validated by in vitro studies, we produced a lung bacterial infection model to further validate the in vivo activity of this combination. Our results indicate that PMNB and GML can significantly increase survival and reduce the bacterial burden in the lung tissue compared with PMNB monotherapy. Conversely, body weights were nonsignificantly increased after combination treatment. The monotherapy of GML had no influence on the bacterial infection treatment. This revealed that the PMNB and GML combination treatment could rescue *K. pneumoniae*-infected mice by accelerating the clearance of bacteria and reducing mortality.

In this study, we confirmed that PMNB synergized with GML increased antimicrobial activity against Gram-negative strains in vitro and in vivo, and this synergistic activity was mainly achieved by exacerbating the destruction of bacterial cell structures, eradicating preformed biofilms, and promoting intracellular ROS accumulation. Our data also highlighted that this combination treatment could be used as a potential therapeutic method for lung bacterial infections. Future studies could focus on bacterial pneumonia caused by other pathogens (*P. aeruginosa* or *A. baumannii*) that play an important role in the prevention of clinical progression and the spread of resistance. Overall, we have indicated that PMNB and GML act synergistically to kill Gram-negative strains, including MDR and PR isolates. Remarkably, the in vitro antimicrobial potency could be well-translated in vivo for PMNB and GML combination treatment. Our work clearly emphasizes the PMNB and GML combination-induced synergistic antibacterial effect and opens the possibility for a new path to treat infections caused by Gram-negative pathogens, particularly those that are MDR pathogens.

## 4. Materials and Methods

### 4.1. Bacterial Isolates, Drugs, and Mice

*Escherichia coli* K-12 strain BW25113, *Acinetobacter baumannii* ATCC 17978, *Klebsiella pneumoniae* ATCC 43816 (obtained from the American Type Culture Collection), and *Pseudomonas aeruginosa* PA14 were wild-type standard strains [35,36,37]. *A. baumannii* HR13 and *K. pneumoniae* KP024 were wild-type clinical MDR pathogens and isolated from ICU patients at a tertiary care hospital in Anhui, China. All isolates were obtained from the Anhui Center for the Surveillance of Bacterial Resistance (Hefei, China) and stored in cation-adjusted Muller–Hinton Broth (CAMHB; Sigma-Aldrich, Burlington, MA, USA) with 50% glycerol in a cryopreservation tube for storage at −80 °C. Bacteria were grown in CAMHB or on CAMHB agar at 37 °C. PMNB was ordered from Sigma-Aldrich (St. Louis, MO, USA). GML was ordered from Solarbio (Shanghai, China). Other reagents were ordered from Sigma-Aldrich. Wild-type C57BL/6 mice were purchased from the Experimental Animal Center of Anhui Province (Hefei, China). All experiments involving mice were approved by the Institutional Animal Care and Use Committee of Anhui Medical University (Hefei, China).

### 4.2. Antimicrobial Susceptibility Testing

Minimum inhibitory concentration (MIC) values were determined through the broth microdilution method. The test medium was CAMHB, and the bacterial culture concentrations were adjusted to 5 × 10^5^ CFU/mL. Various doses of PMNB or GML were mixed and incubated overnight. As previously reported, the MIC was defined as the lowest concentration of antibiotics with no visible bacteria growth by using the Clinical and Laboratory Standards Institute (CLSI) and the European Committee on Antimicrobial Susceptibility Testing (EUCAST) [38,39,40].

### 4.3. Checkerboard Assays

Synergy in vitro was evaluated by checkerboard assay. Bacterial density was diluted and adjusted to ~5 × 10^5^ CFU/mL. The flat-bottomed 96-well plates were set up with serial doubling dilutions of PMNB, GML or in combination at various concentrations. The microwell plates were incubated at 37 °C overnight before reading.

The fractional inhibitory concentration index (FICI) was defined as follows: FICI = FIC_A_ + FIC_B_ = (MIC of drug_A_ in combination/MIC of drug_A_ alone) + (MIC of drug_B_ in combination/MIC of drug_B_ alone). Synergy was defined when FICI ≤ 0.5, additive was defined when 0.5  <  FICI  ≤  1, indifference was defined when FICI > 1 and < 4, and antagonism was defined when FICI ≥ 4 [41]. All determinations were performed at least in triplicate on different days. Wells without drugs served as growth-negative controls, and wells with medium served as background negative controls.

### 4.4. Time–Kill Studies

Time–kill studies were conducted as previously described [42]. Bacterial strains were diluted to 5 × 10^5^ CFU/mL and incubated with PMNB or GML for different time periods. For wild-type standard strains, cultures were treated with PMNB (2× MIC or 4× MIC), GML (500 μg/mL), or PMNB in combination with GML. For MDR clinical strains, PMNB (1× MIC or 2× MIC) alone, GML (500 μg/mL) alone, or both in combination were added due to resistance to PMNB with a high MIC value. At each time point, samples and untreated cultures were collected, diluted, and plated onto agar plates at 37 °C. The next day, the viable colony cell counts (CFU/mL) were measured from the colonies growing on the plates after being left overnight. Controls (untreated) were set as the negative control.

### 4.5. Cellular Morphology Study

A scanning electron microscope (SEM) was used to visualize alterations in the cellular morphology of Gram-negative bacteria. Tested strains were treated with PMNB, GML alone, or in combination, or in broth for 1 h. Briefly, cells were washed 3 times in 1× phosphate-buffered saline (PBS), then grown on cover slips and fixed with 2.5% glutaraldehyde at 37 °C. After washing in PBS, samples were dried through an ascending ethanol series by critical point drying. The samples were observed with a scanning electron microscope (SEM).

### 4.6. Crystal Violet Biofilm Assay

Biofilm formation was investigated by a crystal violet assay as previously described [43]. Briefly, overnight cultures were standardized to exponential phase OD_600_ ~0.3, and then diluted 200-fold in fresh broth. Cultures were inoculated in 96-well polystyrene microtiter plates and co-cultured at 37 °C for 24 h. Following incubation, the biofilms in each well of the plates were carefully washed with sterile PBS to remove unattached cells, then stained with 0.1 % crystal violet solution for 20 min after drying. The wells were then washed and dried as described above, and the stained biofilms were then dried for 1 h and extracted with 33% glacial acetic acid. A volume of 200 μL of the sample solution was measured at 595 nm using a Tecan Spark Multimode Microplate Reader (Tecan, Mannedorf, Switzerland). The percentage of biofilm eradication and inhibition was determined according to the following equation:

The rate of biofilm eradication/inhibition: (%) = (1 − OD_test_/OD_control_) × 100%

### 4.7. ROS Assay

The ROS levels were assessed using a flow cytometer (Beckman Coulter CyAn ADP analyser; Brea, CA, USA) following treatment with a fluorescent probe, such as carboxy-H2DCFDA (Thermo Fisher, Waltham, MA, USA) [42]. Exponentially growing cultures of BW25113 or ATCC 43816 were treated with PMNB (4× MIC) alone, GML (500 μg/mL) alone, or in combination for 15 min. Carboxy-H2DCFDA (final concentration 10 µM) was added to the prepared samples to detect the levels of intracellular ROS. All culture tubes were wrapped with aluminium foil to avoid light. Samples (200 µL) taken at various times were washed with precooled PBS by centrifugation to scavenge the agents and analysed by flow cytometry. A total of 100,000 cells were recorded for each sample to determine fluorescence.

Next, we tested whether ROS-mediated rapid killing is an important mechanism of the PMNB–GML combination. An exogenous ROS scavenger, dimethyl sulfoxide (DMSO), effectively mitigated the ROS levels. Treated samples were placed on agar plates with DMSO (5% v/v), as in the previously described time–kill assay. Growth kinetics were also studied in DMSO alone (Appendix A).

### 4.8. Murine Lung Infection Model

Lung bacterial infection models were constructed to assess the effect of the PMNB and GML combination treatment on antibacterial activity in vivo as previously described [44,45]. C57BL/6 mice (aged 6 weeks; 8 per group) were anaesthetized with isoflurane and inoculated intranasally with 1 × 10^5^ CFU/50 μL of *K. pneumoniae*. Treatment was started 24 h post-infection with PMNB (nebulization) and GML (oral gavage) alone or in combination. Alone, 7.5 mg/kg/day of PMNB and 6 mg/mouse/day of PMNB were used every 12 h [46,47].

The survival and body weights of mice were monitored to evaluate whether the PMNB–GML combination had a therapeutic effect against lung infection. Mice were euthanized by CO_2_ asphyxiation on the third day of treatment, and the lung tissue was removed for later analysis. To assess the CFU counts, lungs were weighed and processed with a tissue homogenizer for serial dilution. Then, samples were plated onto broth agar and colonies were counted after incubation overnight at 37 °C. A blank control (no treatment, PBS) group was set for each condition.

### 4.9. Statistical Analysis

To evaluate statistical differences between groups, unpaired Student’s *t* tests for two groups and one- or two-way analyses of variance (ANOVAs) for multiple groups were performed after testing, with all data points showing a normal distribution. The mouse survival curves were plotted with the Kaplan−Meier method and compared with log-rank tests. A *p* value of < 0.05 was considered statistically significant. All graphs were generated using GraphPad Prism 8.0 (GraphPad Inc., San Diego, CA, USA), FlowJo version 10.4 (Ashland, OR, USA), and Adobe Illustrator CC 2018 (Adobe Systems Inc., Mountain View, CA, USA). All data are representative of independent experiments performed in triplicate, and error bars represent the standard errors of the means (SEM).

## 5. Conclusions

The PMNB and GML combination displayed a synergistic antimicrobial effect on Gram-negative pathogens in vitro and in vivo by exacerbating the damage to the bacterial cell structure, inhibiting biofilm generation, and enhancing ROS generation. Our findings provide a potential therapeutic option to address the prevalent infections caused by MDR Gram-negative pathogens worldwide.

## Figures and Tables

**Figure 1 pathogens-11-00874-f001:**
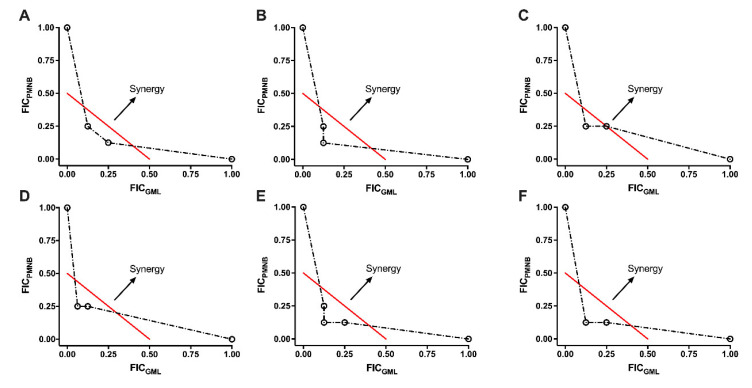
Synergistic effect between PMNB and GML against Gram-negative strains. Isobolograms of the combination of PMNB and GML against (**A**) *E. coli* BW25113, (**B**) *K. pneumoniae* ATCC 43816, (**C**) *A. baumannii* ATCC 17978, (**D**) *P. aeruginosa* PA14, (**E**) *K. pneumoniae* KP024, and (**F**) *A. baumannii HR13*. The red line indicates the ideal isobole; data points below or on this line are evidence of synergy. PMNB, polymyxin B; GML, glycerol monolaurate; FICI, fractional inhibitory concentration index. Data represent three independent biological replicates.

**Figure 2 pathogens-11-00874-f002:**
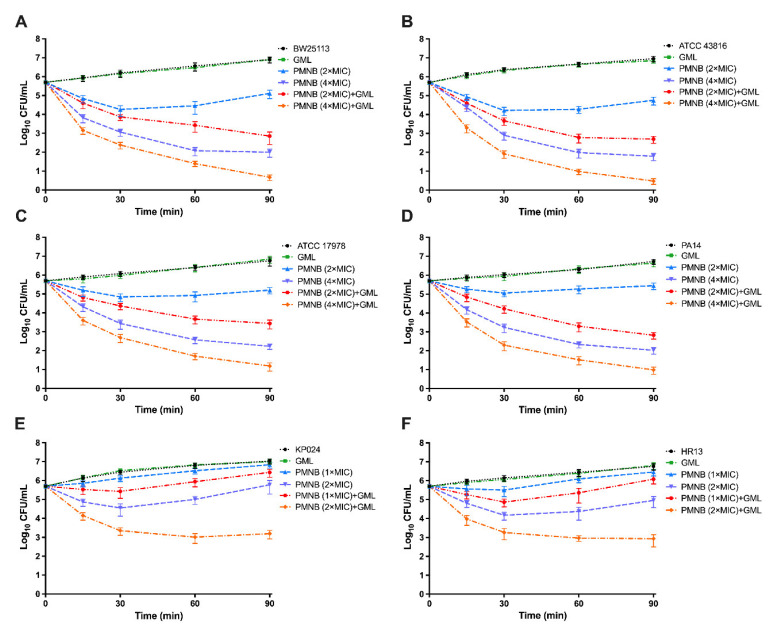
Killing kinetics of PMNB in combination with GML against Gram-negative strains. Time–kill curves for polymyxin B (PMNB) or glycerol monolaurate (GML) monotherapy or their combination therapy against Gram-negative strains within 90 min. (**A**) *E. coli* BW25113, (**B**) *K. pneumoniae* ATCC 43816, (**C**) *A. baumannii* ATCC 17978, and (**D**) *P. aeruginosa* PA14 exposed to 2× or 4× MIC PMNB, GML (500 μg/mL), and in combination (**E**) *K. pneumoniae* KP024 and (**F**) *A. baumannii* HR13 exposed to 1× or 2× MIC PMNB, GML (500 μg/mL), and in combination. Data represent three independent biological replicates; error bars indicate SEM (standard errors of the means).

**Figure 3 pathogens-11-00874-f003:**
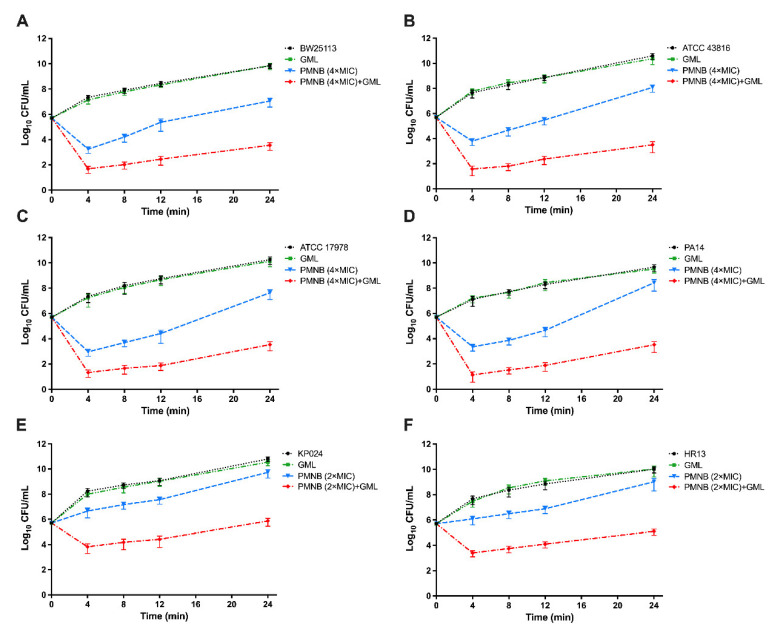
The 24 h kill-kinetics of PMNB and GML combination therapy against Gram-negative bacterial strains. (**A**) *E. coli* BW25113, (**B**) *K. pneumoniae* ATCC 43816, (**C**) *A. baumannii* ATCC 17978, and (**D**) *P. aeruginosa* PA14 exposed to 4× MIC PMNB, GML (500 μg/mL), and in combination. (**E**) *K. pneumoniae* KP024 and (**F**) *A. baumannii* HR13 exposed to 2× MIC PMNB, GML (500 μg/mL), and in combination. Data represent three independent biological replicates; error bars indicate SEM (standard errors of the means).

**Figure 4 pathogens-11-00874-f004:**
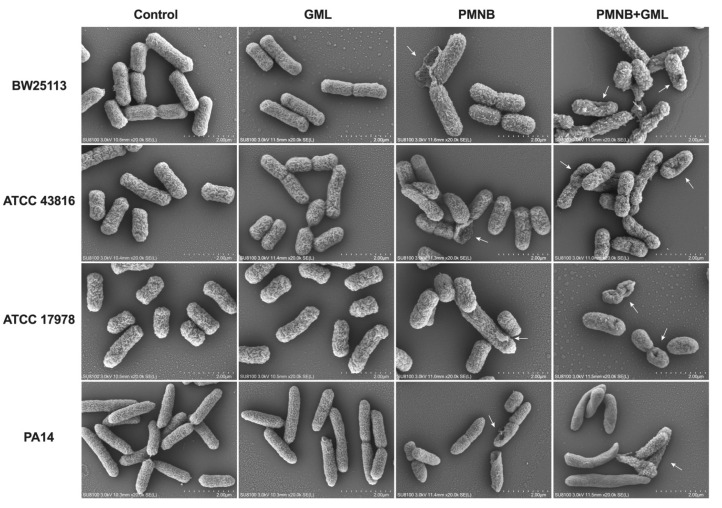
Impact of PMNB and GML on bacterial morphology. Scanning electron micrograph (SEM) images of *E. coli* BW25113, *K. pneumoniae* ATCC 43816, *A. baumannii* ATCC 17978, and *P. aeruginosa* PA14 after treatment with PMNB alone, GML alone, or both. Scale bar: 2 μm. White arrows show morphological damage in bacterial cells. Controls, no treatment; PMNB, polymyxin B; GML, glycerol monolaurate.

**Figure 5 pathogens-11-00874-f005:**
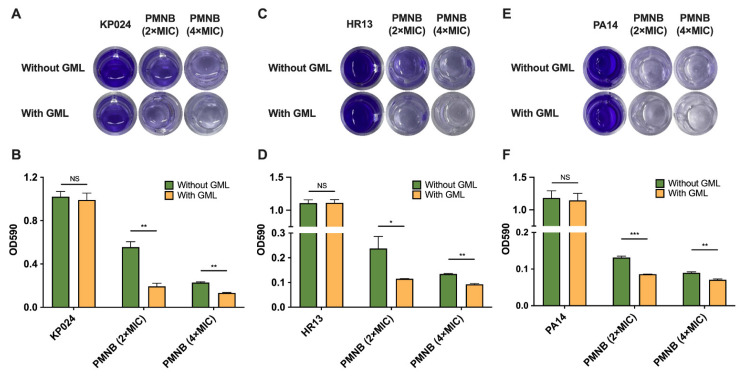
Inhibitory effects of PMNB and GML on biofilm formation. Crystal violet staining assay of biofilms for (**A**) *K. pneumoniae* KP024, biomass quantification of biofilms of (**B**) *K. pneumoniae* KP024, (**C**) *A. baumannii* H, (**D**) *A. baumannii* HR13, (**E**) *P. aeruginosa* PA14 after treatment and (**F**) *P. aeruginosa* PA14 by detecting at OD_590_ (the optical density). *, *p* < 0.05; ** *p* < 0.01; ***, *p* < 0.001; NS, no significance. PMNB, polymyxin B; GML, glycerol monolaurate. Error bars represent SEM (standard errors of the means).

**Figure 6 pathogens-11-00874-f006:**
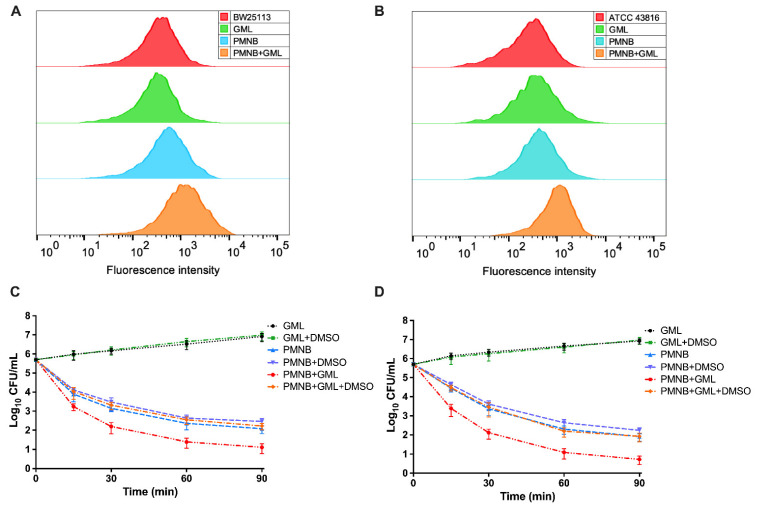
ROS contribute to the PMNB–GML combination-induced lethality. (**A**) *E. coli* BW25113 and (**B**) *K. pneumoniae* ATCC 43816 were co-incubated with carboxy-H2DCFDA (10 mM) for 15 min before detection. (**C**,**D**) 5% DMSO was added to remove ROS accumulation. Combination-induced killing was attenuated in the presence of DMSO for (**C**) BW25113 and (**D**) ATCC 43816 compared with no DMSO at 90 min. PMNB, polymyxin B; GML, glycerol monolaurate. All results were from at least three independent repetitions; error bars represent SEM (standard errors of the means).

**Figure 7 pathogens-11-00874-f007:**
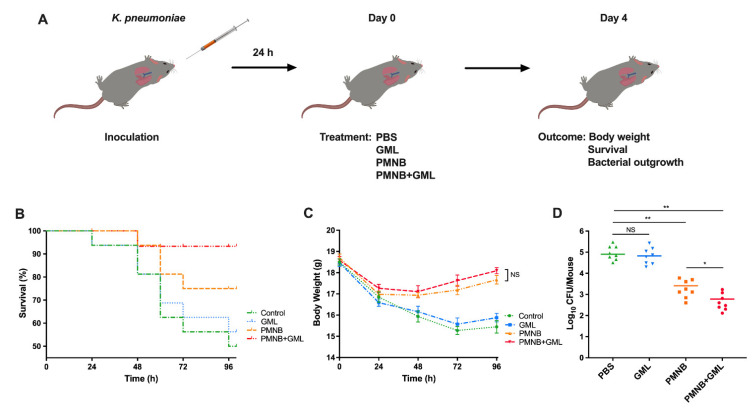
Effect of PMNB and GML combination on lung infection caused by *K. pneumoniae*. (**A**) Schematic diagram of lung infection caused by *K. pneumoniae* ATCC 43816 and the treatment process. (**B**) Survival rates and (**C**) body weight of infected mice were recorded. (**D**) Pulmonary bacterial burdens in treatment with PBS, PMNB, GML, and in combination at 3 days. The group size was 8 mice. *, *p* < 0.05; **, *p* < 0.01; NS, no significance; PMNB, polymyxin B; GML, glycerol monolaurate. One experiment is representative of three independent biological replicates.

**Table 1 pathogens-11-00874-t001:** Antimicrobial susceptibility and combined effects of PMNB and GML alone or in combination against Gram-negative strains.

Bacterial Isolate	Agent	MIC (μg/mL)		MIC_In combination_/MIC_singly_	FICI	Outcome
		Singly	In Combination			
*E. coli* BW25113	PMNB	0.5	0.125	0.25	<0.375	Synergy
	GML	>4000	500	<0.125		
*K. pneumoniae* ATCC 43816	PMNB	1	0.125	0.125	<0.25	Synergy
	GML	>4000	500	<0.125		
*K. pneumoniae* KP024	PMNB	8	1	0.125	<0.25	Synergy
	GML	>4000	500	<0.125		
*A. baumannii* ATCC 17978	PMNB	0.5	0.125	0.25	<0.375	Synergy
	GML	>4000	500	<0.125		
*A. baumannii* HR13	PMNB	4	0.5	0.125	<0.25	Synergy
	GML	>4000	500	<0.125		
*P. aeruginosa* PA14	PMNB	0.5	0.125	0.25	<0.3125	Synergy
	GML	>4000	250	<0.0625		

MIC, minimum inhibitory concentration; PMNB, polymyxin B; GML, glycerol monolaurate; FICI, fractional inhibitory concentration index. Data represent three independent biological replicates.

## Data Availability

Data are contained within the article and Appendix A.

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
