# Peer review of "Polymyxin B in Combination with Glycerol Monolaurate Exerts Synergistic Killing against Gram-Negative Pathogens"

_pathogens, 2022, doi:10.3390/pathogens11080874_

Round 1

Reviewer 1 Report

The manuscript pathogens-1800902 Polymyxin B in combination with glycerol monolaurate exerts synergistic killing against Gram-negative pathogens by Yun Zheng et al. describes the a study of the co-pharmacological effect of polymyxin B and glycerol monolaurate. The topic of the study is indeed important and interesting because of the global increase in drug resistance of microorganisms.

The manuscript is logical and well written. The prospects for the practical use of the results obtained are high. The paper will definitely be of interest to the readers of Pathogens.

Questions and comments:

1) Line 102 - Why do you conclude that the bacteria are resistant to polymyxin B at MIC values ≥ 4? A relevant reference is necessary. 

2) Figure 1 is more appropriate in Section 2.2.

3) Line 421 - Title section 4.8 Murine skin infection model should certainly be replaced by Murine lung infection model

4)  The authors demonstrated a synergistic effect and a decrease of  polymyxin MIC in the presence of glycerol monolaurate. However, it is equally important to discuss the reduced nephrotoxicity of the bicomponent formulation compared with pure polymyxin B.

5) Improving the safety profile of polymyxins is a top challenge in pharmaceutical R&D. Perhaps some publications will be of interest to you:

  • W.H. Organization, WHO publishes list of bacteria for which new antibiotics are urgently needed,  (2017)

  • doi.org/10.1016/j.ijbiomac.2021.07.114

  • doi.org/10.3390/ijms22168381

  • 10.1016/j.ijbiomac.2022.06.080

  • 10.1016/j.ijantimicag.2018.09.003

  • doi.org/10.3390/ijms23052771

Author Response

Dear Prof. Zhang and Reviewer,

I’m Jiabin Li, the corresponding author of Manuscript ID: pathogens-1800902. We appreciate the editors and reviewers’ interests and positive comments on our manuscript. We are grateful for your constructive comments to improve the quality of this manuscript.

We have revised this manuscript and made significant changes to meet the high standard of the journal (the modification to reduce duplication are annotated in blue; respond to comments are annotated in orange). Following is the point-by-point responses to the comments from the editor and reviewers. Hopefully we will address your concerns!

Prof, Jiabin Li, PhD.

Reviewer 1:

The manuscript pathogens-1800902 Polymyxin B in combination with glycerol monolaurate exerts synergistic killing against Gram-negative pathogens by Yun Zheng et al. describes the study of the co-pharmacological effect of polymyxin B and glycerol monolaurate. The topic of the study is indeed important and interesting because of the global increase in drug resistance of microorganisms.

The manuscript is logical and well written. The prospects for the practical use of the results obtained are high. The paper will definitely be of interest to the readers of Pathogens.

Questions and comments:

1) Line 102 - Why do you conclude that the bacteria are resistant to polymyxin B at MIC values ≥ 4? A relevant reference is necessary.

Response: We appreciate the good question raised by the reviewer.

As previously depicted [1], polymyxin B MIC is ≤ 2 mg/L for polymyxin-susceptible A. baumannii, ≥ 4 mg/L for polymyxin-resistant A. baumannii according to the CLSI guidelines. For K. pneumoniae, susceptibility to polymyxin B was extrapolated from colistin breakpoints where susceptibility is defined as an MIC ≤2 mg/L and resistance an MIC of >2 mg/L according to the European Committee on Antimicrobial Susceptibility Testing (EUCAST).

In our study, the MIC for polymyxin B with A. baumannii HR13 was 4 μg/mL, and the MIC for polymyxin B with K. pneumoniae KP024 was 8 μg/mL. Thus, we summarize them as MIC values ≥ 4, and these two strains resistant to polymyxin B.

We have revised this description, “two clinical MDR isolates A. baumannii HR13 (MIC, 4 μg/mL) and K. pneumoniae KP024 (MIC, 8 μg/mL) were resistant to PMNB” in lines 109-111. And modified Section 4.2 “As previously reported, MIC was defined as the lowest concentration of antibiotic with no visible bacteria growth by using the Clinical and Laboratory Standards Institute (CLSI) and the European Committee on Antimicrobial Susceptibility Testing (EUCAST).” in lines 374-377.

A relevant reference is cited in Reference 40 (line 589-591):

[40] Tran TB, Bergen PJ, Creek DJ, Velkov T, Li J. Novel polymyxin combination with antineoplastic mitotane improved the bacterial killing against polymyxin-resistant multidrug-resistant Gram-negative pathogens. Front Microbiol. 2018. 9:721. doi: 10.3389/fmicb.2018.00721.

2) Figure 1 is more appropriate in Section 2.2.

Response: We appreciate the reviewer’s constructive suggestion. We have adjusted the position of Figure 1 in Section 2.2 to achieve compact representations in typography. Please see lines 118.

3) Line 421 - Title section 4.8 Murine skin infection model should certainly be replaced by Murine lung infection model

Response: Thank you for the positive comments. We have corrected this description. Please see line 436.

4) The authors demonstrated a synergistic effect and a decrease of polymyxin MIC in the presence of glycerol monolaurate. However, it is equally important to discuss the reduced nephrotoxicity of the bicomponent formulation compared with pure polymyxin B.

Response: We thank the reviewer for the good comments, which have been taken into careful consideration.

Antimicrobial combination are promising ways to solve this problem by achieving the desired therapeutic effect at a lower concentration and reducing side effects such as nephrotoxicity. We have added this sentence “and achieving the desired therapeutic effect at a lower concentration to dilute serious side effects such as nephrotoxicity” to emphasize the importance of this problem, please see lines 87-89.

We absolutely agree with the reviewer. Meanwhile, we will continue to study its effect for clinical strains and in vivo possible toxicity of PMNB-GML combination, then to report our latest results as we are finding out more.

5) Improving the safety profile of polymyxins is a top challenge in pharmaceutical R&D. Perhaps some publications will be of interest to you:

W.H. Organization, WHO publishes list of bacteria for which new antibiotics are urgently needed,  (2017)

doi.org/10.1016/j.ijbiomac.2021.07.114

doi.org/10.3390/ijms22168381

10.1016/j.ijbiomac.2022.06.080

10.1016/j.ijantimicag.2018.09.003

doi.org/10.3390/ijms2305277

Response: We appreciate the reviewer’s kind comments. These publications are great and give us important information after reading and thinking. We are very interested in them and thank you again. We have added the related statement in Introduction “Several recent publications demonstrated that it is very important to control the growth of MDR Gram-negative pathogens, increase local bioavailability and reduce systemic tox-icity through improving the therapeutic characteristics of polymyxin, such as nano-technology-based modification and combination therapy.” (lines 70-73) and mostly considered and cited four literatures in the References 9-12 (lines 514-525) as follows:

  • Dubashynskaya NV, Raik SV, Dubrovskii YA, Shcherbakova ES, Demyanova EV, Shasherina AY, Anufrikov YA, Poshina DN, Dobrodumov AV, Skorik YA. Hyaluronan/colistin polyelectrolyte complexes: Promising antiinfective drug delivery systems. Int J Biol Macromol. 2021. 30;187:157-165. doi: 10.1016/j.ijbiomac.2021.07.114.
  • Dubashynskaya NV, Bokatyi AN, Gasilova ER, Dobrodumov AV, Dubrovskii YA, Knyazeva ES, Nashchekina YA, Demyanova EV, Skorik YA. Hyaluronan-colistin conjugates: Synthesis, characterization, and prospects for medical applications. Int J Biol Macromol. 2022. 17;215:243-252. doi: 10.1016/j.ijbiomac.2022.06.080.
  • Dubashynskaya NV, Raik SV, Dubrovskii YA, Demyanova EV, Shcherbakova ES, Poshina DN, Shasherina AY, Anufrikov YA, Skorik YA. Hyaluronan/diethylaminoethyl chitosan polyelectrolyte complexes as carriers for improved colistin delivery. Int J Mol Sci. 2021. 4;22(16):8381. doi: 10.3390/ijms22168381.
  • Otto RG, van Gorp E, Kloezen W, Meletiadis J, van den Berg S, Mouton JW. An alternative strategy for combination therapy: Interactions between polymyxin B and non-antibiotics. Int J Antimicrob Agents. 2019. 53(1):34-39. doi: 10.1016/j.ijantimicag.2018.09.003.

Reviewer 2 Report

kThis is a very well-designed work on the recent advancement of antimicrobial drug discovery. My individual comments are below 

.      Page 2 Line 56 bacterial name spelling recheck

2.       Page 2 line 58 “gallium” should be replaced by “Calcium”

3.       Page 2 line 91-93 please complete the sentence

4.       Table 1 page 3: I would like to see the FIC data of a pseudomonas species that is resistant to polymyxin (MIC 2µg/mL). As we are talking about polymyxin resistance as a background, you should choose isolates that are especially resistant to polymyxin.

5.       Figure 2 page 4: 90 min of time-kill data for a synergistic killing of gram-negative bacteria is insufficient for making credible conclusion. I suggest making at least several other time points such as 4h, 8h, 12h

6.       Figure 2: Bacterial time-kill should start from Log10 CFU/mL value of 8

7.       Page 11 line 395: OD600 (600 should be subscript)

8.       Page 11 line 405: rewrite the equation with correct fonts and subscript

9.       Line 430 (CO2) should be written properly putting 2 as a subscript

Author Response

Dear Prof. Zhang and Reviewer,

I’m Jiabin Li, the corresponding author of Manuscript ID: pathogens-1800902. We appreciate the editors and reviewers’ interests and positive comments on our manuscript. We are grateful for your constructive comments to improve the quality of this manuscript.

We have revised this manuscript and made significant changes to meet the high standard of the journal (the modification to reduce duplication are annotated in blue; respond to comments are annotated in orange). Following is the point-by-point responses to the comments from the editor and reviewers. Hopefully we will address your concerns!

Prof, Jiabin Li, PhD.

Reviewer 2:

This is a very well-designed work on the recent advancement of antimicrobial drug discovery. My individual comments are below

  1. Page 2 Line 56 bacterial name spelling recheck

Response: We are thankful for the reviewer pointing out this problem. We have rechecked and corrected this bacterial name spelling “Paenibacillus polymyxa”, please see line 56.

  1. Page 2 line 58 “gallium” should be replaced by “Calcium”

Response: Thank you for the positive comments. We have corrected this mistake, please see line 58.

  1. Page 2 line 91-93 please complete the sentence.

Response: We appreciate the reviewer’s constructive suggestion. We have completed and revised this sentence “We demonstrated that ROS generation, bacterial cell membrane integrity disruption, and biofilm inhibition are critical elements in synergistic antimicrobial activity by using combining scanning electron microscope (SEM), biofilm formation assay and ROS assay.” Please see line 97-100.

  1. Table 1 page 3: I would like to see the FIC data of a pseudomonas species that is resistant to polymyxin (MIC ≥ 2 µg/mL). As we are talking about polymyxin resistance as a background, you should choose isolates that are especially resistant to polymyxin.

Response: We thank the reviewer for the good comments, which have been taken into careful consideration.

We therefore screened our library of Pseudomonas aeruginosa strains and found the MICs of two clinical isolates (P. aeruginosa NY04 and P. aeruginosa NY06) to PMNB is 2 µg/mL. Then, we have further investigated the FIC results of these two clinical isolates in PMMB and GML combination. The results are depicted in following table.

Bacterial isolate

Agent

MIC (μg/mL)

MICIn combination/MICsingly

FICI

Outcome

Singly

In combination

P. aeruginosa NY04

PMNB

2

0.5

0.25

< 0.375

Synergy

GML

> 4000

500

< 0.125

P. aeruginosa NY06

PMNB

2

0.25

0.125

< 0.375

Synergy

GML

> 4000

1000

< 0.25

This combination treatment yielded a much better therapeutic effect than PMNB alone, and exhibited significant biofilm inhibition effect, especially P. aeruginosa. Considering that the ability to form biofilms contributes significantly to the clinical burden of P. aeruginosa infection, we next will explore the mechanisms of this combination inhibition of biofilms and search for unique molecular targets in P. aeruginosa.

  1. Figure 2 page 4: 90 min of time-kill data for a synergistic killing of gram-negative bacteria is insufficient for making credible conclusion. I suggest making at least several other time points such as 4h, 8h, 12h

Response: We appreciate the reviewer’s constructive suggestion, which improve the quality of the paper.

We have completed the time-killing kinetics experiments at 4h, 8h, 12h and 24h. The results are plotted in Figure 3 and the related descriptions were also added in the revised manuscript.

Please see section 2.3:

“Therefore, we further used 4× MIC PMNB to test the synergistic killing with or without GML within 24 h and found a stable antibacterial activity (Figure 3). This combination treatment resulted in strong killing ~3.8 log10 CFU/mL killing of E. coli BW25113 (Figure 3A), ~4.5 log10 CFU/mL killing of K. pneumoniae ATCC 43816 (Figure 3B), ~4.1 log10 CFU/mL killing of A. baumannii ATCC 17978 (Figure 3C) and ~4.8 log10 CFU/mL killing of P. aeruginosa PA14 (Figure 3D) compared to PMNB alone at 24 h.” in lines 154-159;

“Figure 3. The 24 h-kill kinetics of PMNB and GML combination therapy against Gram-negative bacterial strains. (A) E. coli BW25113, (B) K. pneumoniae ATCC 43816, (C) A. baumannii ATCC 17978 and (D) P. aeruginosa PA14 exposed to 4× MIC PMNB, GML (500 μg/mL), and in combination, respectively. (E) K. pneumoniae KP024 and (F) A. baumannii HR13 exposed to 2× MIC PMNB, GML (500 μg/mL), and in combination, respectively. Data represent three independent biological replicates; error bars indicate SEM (standard errors of the means).” in lines 160-166;

and “Similarly, PMNB and GML combination therapy showed powerful bactericidal effect with ~4 log10 CFU/mL killing for KP024 (Figure 3E) and ~3.7 log10 CFU/mL killing for HR13 (Figure 3F) following 24 h incubation, when compared with PMNB alone.” in lines 171-174.

Please see section 3 “rapid and stable synergistic killing” in line 283 and section 4.4 “incubated with PMNB or GML for different time points” in lines 392.

  1. Figure 2: Bacterial time-kill should start from Log10 CFU/mL value of 8

Response: We appreciate the reviewer’s positive comments and also agree with the reviewer.

In our methods, overnight cultures were diluted and grown to exponential phase OD600 ~0.3, then the cultures were diluted in broth to a final bacterial inoculum of 5×105 CFU/mL, which based on previous experiments reported in literatures[1-5]. Because of this fact, our results of time-kill can be persuasive.

  • Tran TB, SE Cheah, HH Yu, et al. Anthelmintic closantel enhances bacterial killing of polymyxin B against multidrug-resistant Acinetobacter baumannii. J Antibiot. 2016. 69:415–421.
  • Liu X, Sun X, Deng X, et al. Calycosin enhances the bactericidal efficacy of polymyxin B by inhibiting MCR-1 in vitro. J Appl Microbiol. 2020. 129(3):532-540. doi: 10.1111/jam.14635.
  • Bowers DR, Cao H, Zhou J, et al. Assessment of Minocycline and Polymyxin B Combination against Acinetobacter baumannii. Antimicrob Agents Chemother. 2015. 59(5): 2720–2725. doi: 10.1128/AAC.04110-14.
  • Gaurav A, Kothari A, Omar BJ, Pathania R. Assessment of polymyxin B–doxycycline in combination against Pseudomonas aeruginosa in vitro and in a mouse model of acute pneumonia. Int J Antimicrob Agents. 2020 Jul;56(1):106022. doi: 10.1016/j.ijantimicag.2020.106022.
  • Borjan J, Meyer KA, Shields RK, Wenzler E. Activity of ceftazidime-avibactam alone and in combination with polymyxin B against carbapenem-resistant Klebsiella pneumoniae in a tandem in vitro time-kill/in vivo Galleria mellonella survival model analysis. Int J Antimicrob Agents. 2020. 55(1):105852. doi: 10.1016/j.ijantimicag.2019.11.009.

  1. Page 11 line 395: OD600 (600 should be subscript)

Response: Thank you for the positive comments. We have revised this written format “OD600”, please see line 410.

  1. Page 11 line 405: rewrite the equation with correct fonts and subscript

Response: Thank you for the reviewer pointing out this problem. We have modified this equation “The rate of biofilm eradication/inhibition (%) = (1 – ODtest / ODcontrol) × 100 %”, please see line 420.

  1. Line 430 (CO2) should be written properly putting 2 as a subscript

Response: Thank you for the positive comments. We have revised this written format “CO2”, please see line 445.

Reviewer 3 Report

Zheng et al reported a promising combination of polymyxin B with glycerol monolaurate against Gram-negative pathogens. This is a very timely topic as no novel classes of antibiotics will be available for this life-threatening ‘superbug’ for many years to come, it is very important to pursue every innovative approach to rescue the currently available antibiotics on the market. The author evaluated the synergistic efficacy of this combination from in vitro to in vivo. However, in my point of view, the current manuscipt should be improved in the following parts:

(1) Based on the pk of polymyxin B, I do not believe the efficacy of polymyxin B monotherapy and in combination from the mouse lung infection model. The dose of polymyxin B in 3 mg/kg/day is too low as polymyxins have poor lung penetration following such IP or IV (Landersdorfer CB, et al Journal of Antimicrobial Chemotherapy,73(2), pp.462-468.). The concentrations achieved in lung were likely very low and won't have significant killing effect. CLSI also mentioned "When colistin or polymyxin B is given systemically, the drug is unlikely to be effective for pneumonia". The author will have measure the exact concentration of polymyxin B in lung to prove their results and conclusion. 

(2) Since there is not breakpoint for GML. I don't think the author can draw the conclusion that all strains they used in the current study were more sensitive to PMB than GML. 

(3) Please double check the captions for Figure 4. 

Author Response

Dear Prof. Zhang and Reviewer,

I’m Jiabin Li, the corresponding author of Manuscript ID: pathogens-1800902. We appreciate the editors and reviewers’ interests and positive comments on our manuscript. We are grateful for your constructive comments to improve the quality of this manuscript.

We have revised this manuscript and made significant changes to meet the high standard of the journal (the modification to reduce duplication are annotated in blue; respond to comments are annotated in orange). Following is the point-by-point responses to the comments from the editor and reviewers. Hopefully we will address your concerns!

Prof, Jiabin Li, PhD.

Reviewer 3:

Zheng et al reported a promising combination of polymyxin B with glycerol monolaurate against Gram-negative pathogens. This is a very timely topic as no novel classes of antibiotics will be available for this life-threatening ‘superbug’ for many years to come, it is very important to pursue every innovative approach to rescue the currently available antibiotics on the market. The author evaluated the synergistic efficacy of this combination from in vitro to in vivo. However, in my point of view, the current manuscipt should be improved in the following parts:

(1) Based on the pk of polymyxin B, I do not believe the efficacy of polymyxin B monotherapy and in combination from the mouse lung infection model. The dose of polymyxin B in 3 mg/kg/day is too low as polymyxins have poor lung penetration following such IP or IV (Landersdorfer CB, et al Journal of Antimicrobial Chemotherapy,73(2), pp.462-468.). The concentrations achieved in lung were likely very low and won't have significant killing effect. CLSI also mentioned "When colistin or polymyxin B is given systemically, the drug is unlikely to be effective for pneumonia". The author will have measure the exact concentration of polymyxin B in lung to prove their results and conclusion.

Response: We appreciate the reviewer’s constructive suggestion. We also completely agree with the reviewer and have previously considered this issue ourselves.

In the course of the previous experiment, due to our results showed that PMNB-GML combination possess the synergistic antibacterial efficacy in vivo, thus a further study was not assessed to measure the actual drug concentration and pharmacodynamics in lung. After serious consideration, we have decided to modify the route of administration of PMNB and reconstruct the in vivo experiments to confirm our conclusion of the present a study. By modifying the in vivo experiments, the synergistic antibacterial efficacy was observed again, thus we verified our conclusion in vivo (Figure 7).

We have added and improved the related statement in the revised manuscript (Please see lines 247-252, 255-256, 440-443 and 605-607).

(2) Since there is not breakpoint for GML. I don't think the author can draw the conclusion that all strains they used in the current study were more sensitive to PMB than GML.

Response: We thank the reviewer’s positive comments and absolutely agree with the reviewer.

In fact, we would like to point out that the antibacterial ability of GML is far less than PMNB, or MIC values of GML are much higher than PMNB MIC values. However, we believe that this statement is superfluous after careful consideration and then we have deleted it.

(3) Please double check the captions for Figure 4.

Response: We are thankful for the reviewer pointing out this deficiency. We have revised this caption for Figure 4 “Inhibitory effect of PMNB and GML on biofilms formation. Crystal violet staining assay of bio-films for (A) K. pneumoniae KP024, (C) A. baumannii HR13 and (E) P. aeruginosa PA14 after treatments. Biomass quantification of biofilms in (B) K. pneumoniae KP024, (D) A. baumannii HR13 and (F) P. aeruginosa PA14 by detecting at OD590 (the optical density).” Please see lines 212-217.

Round 2

Reviewer 1 Report

The article may be published.

Author Response

We appreciate the reviewer’s positive comments, and thank you for your contributions to our article.

Reviewer 3 Report

The revised manuscript improved a lot. I do not have further comments except for the format check for references.

Author Response

Dear Reviewer,

I’m Jiabin Li, the corresponding author of Manuscript ID: pathogens-1800902. We have checked and revised the format for references in this manuscript. We appreciate the reviewer’s positive comments, and thank you for your contributions to our article.

Hopefully we will address your concerns!

Prof, Jiabin Li, PhD.

This manuscript is a resubmission of an earlier submission. The following is a list of the peer review reports and author responses from that submission.